# Development of a High-Power Capacity Open Source Electrical Stimulation System to Enhance Research into FES-Assisted Devices: Validation of FES Cycling

**DOI:** 10.3390/s22020531

**Published:** 2022-01-11

**Authors:** Tiago Coelho-Magalhães, Emerson Fachin-Martins, Andressa Silva, Christine Azevedo Coste, Henrique Resende-Martins

**Affiliations:** 1Graduate Program in Electrical Engineering, Universidade Federal de Minas Gerais, Av. Antônio Carlos 6627, Belo Horizonte 31270-901, Brazil; henriquerm@ufmg.br; 2Plataforma de Serviços Tecnológicos BEMTEVI, Parque Científico e Tecnológico, Universidade de Brasília, Brasília 70910-900, Brazil; efmartins@unb.br; 3Centro de Treinamento Esportivo da Escola de Educação Física, Fisioterapia e Terapia Ocupacional, Universidade Federal de Minas Gerais, Belo Horizonte 31310-000, Brazil; andressa@demello.net.br; 4National Institute for Research in Computer Science and Automation (Inria), Camin Team, 34090 Montpellier, France; christine.azevedo@inria.fr

**Keywords:** functional electrical stimulation, FES cycling, neuromuscular electrical stimulation, FES-assisted cycling

## Abstract

Since the first Cybathlon 2016, when twelve teams competed in the FES bike race, we have witnessed a global effort towards the development of stimulation and control strategies to improve FES-assisted devices, particularly for cycling, as a means to practice a recreational physical activity. As a result, a set of technical notes and research paved the way for many other studies and the potential behind FES-assisted cycling has been consolidated. However, engineering research needs instrumented devices to support novel developments and enable precise assessment. Therefore, some researchers struggle to develop their own FES-assisted devices or find it challenging to implement their instrumentation using commercial devices, which often limits the implementation of advanced control strategies and the possibility to connect different types of sensor. In this regard, we hypothesize that it would be advantageous for some researchers in our community to enjoy access to an entire open-source FES platform that allows different control strategies to be implemented, offers greater adaptability and power capacity than commercial devices, and can be used to assist different functional activities in addition to cycling. Hence, it appears to be of interest to make our proprietary electrical stimulation system an open-source device and to prove its capabilities by addressing all the aspects necessary to implement a FES cycling system. The high-power capacity stimulation device is based on a constant current topology that allows the creation of biphasic electrical pulses with amplitude, width, and frequency up to 150 mA, 1000 µs, and 100 Hz, respectively. A mobile application (Android) was developed to set and modify the stimulation parameters of up to eight stimulation channels. A proportional-integral controller was implemented for cadence tracking with the aim to improve the overall cycling performance. A volunteer with complete paraplegia participated in the functional testing of the system. He was able to cycle indoors for 45 min, accomplish distances of more than 5 km using a passive cycling trainer, and pedal 2400 m overground in 32 min. The results evidenced the capacity of our FES cycling system to be employed as a cycling tool for individuals with spinal cord injury. The methodological strategies used to improve FES efficiency suggest the possibility of maximizing pedaling duration through more advanced control techniques.

## 1. Introduction

Spinal Cord Injury (SCI) may result in complete or incomplete paralysis interfering in the neurological signal transmission and modulation across and below the level of the injury, leading to impairments in autonomic, sensory, and neuromusculoskeletal and movement-related functions [1]. Individuals affected by this injury often experience a considerable reduction in their mobility, resulting in alterations in metabolic functions and body composition [2,3,4]. Long-term complications after SCI include respiratory, cardiovascular, urinary, and bowel dysfunction, in addition to spasticity, pain syndromes, and pressure ulcers [5]. 

Rehabilitation programs may consider Functional Electrical Stimulation (FES) to restore or improve the functional capacity of paretic or paralyzed muscles in this population [6]. FES is a technique in which electrical pulses are applied to the nerve fibers to artificially evoke contractions and produce functional movements such as walking, rowing, and cycling [7,8,9]. Rehabilitation programs assisted by FES have demonstrated impact on pain relief, cardiorespiratory function, body composition, and bone metabolism [10,11,12,13].

FES-assisted cycling (or FES cycling) is a common rehabilitation technique that allows individuals with little or no voluntary leg movement to pedal on a stationary exercise machine or to ride a recumbent tricycle (trike) on a stationary cycling trainer or overground [14]. In general terms, FES-assisted cycling evokes alternate contractions of specific lower limb muscles in a coordinated manner, following a sequence that varies according to the speed and angle of the pedal of the tricycle [15,16]. 

Despite the possible benefits of FES-assisted cycling, the ability to perform cyclic movement depends on numerous physiological variables in response to the electrical stimuli. Unlike the process of voluntary contraction, the electrical stimulation activates motor neurons in a random and synchronized fashion, leading to a reverse recruitment order. Furthermore, the excitability of motor neurons decreases as they are stimulated, which makes it difficult to continue their depolarization through external stimuli [17]. If the intensity of the electrical pulse remains constant, the failure of the excitation–contraction coupling is likely to occur. Added to the non-physiological aspects of motor unit recruitment, this results in greater muscle fatigue and reduced session duration, restricting the benefits offered by the modality and preventing the widespread use of this technique for leisure activities outside of clinics [18].

In response to these restrictions, and in addition to promoting synchronicity with functional activities, electrical stimulation systems are encouraged to be designed with a closed-loop architecture to improve stimuli efficiency, allowing the parameters (frequency, pulse width, and intensity) to be dynamically changed. Furthermore, given the need for pulses to feature higher energy, as deeper fibers are needed for contraction, the stimulation system must feature a high-power capacity in order to support functional activities for longer periods and to enable subjects who are less responsive to stimulation to start functional activities assisted by FES [19,20]

In a previous paper, we detailed a new electrical stimulation system architecture that was first tested for drop-foot correction applications and that demonstrated the ability to be applied to other relevant activities assisted by functional electrical stimulation [21]. In this study, based on the same system topology, we aimed to evaluate stimulation and control strategies for FES-assisted cycling for subjects with SCI. We addressed the functional results achieved by a FES cycling training protocol applied to an individual with paraplegia during a 12-month period. We hypothesized that it was possible to delay the fatigue process and maximize pedaling duration through the closed-loop cadence control. 

Finally, considering that a high-power capacity open-source electrical stimulator device would favor the technological readiness of FES-assisted devices, it is of interest to make available the whole FES-assisted cycling project, including the entire range of hardware, firmware, software, and mechanical designs, since this will allow other researchers to study and modify the project for any purpose. See Appendix A for the Project.

## 2. Materials and Methods

### 2.1. Proprietary Electrostimulation System

Different muscle group combinations can be electrically evoked to perform the cycling movement [22,23,24]. FES cycling strategies usually involve the electrical stimulation of the quadriceps, hamstrings, tibialis anterior, gastrocnemius, and gluteal muscles [25]. As most methodologies suggest the activation of more than two muscle groups bilaterally [26,27], we developed an 8-channel electrostimulation system with a constant current topology to study a broader range of possible muscle combinations for cycling and other modalities. Figure 1 illustrates the system architecture designed.

Although the stimulation channel topology has been discussed in previous work [21], we felt the need to provide detailed information about the electronic design of the stimulation unit. The system can generate symmetric biphasic pulses with a width and frequency up to 1000 µs and 100 Hz, respectively. A DC/DC converter provides high-output voltage and its association with the system topology makes it possible to achieve pulse amplitudes up to 150 mA. A DC/DC module A50-120Y (American Power Design^®^, Windham, NH, USA) was employed in the latest electrostimulation system design, which is available for download. The module A15-150Y was also used and tested.

As Figure 1 suggests, a microcontroller (ARM^®^Crtex-M7) is responsible for the system’s resource management. An external Digital-to-Analog Converter (DAC) generates a signal for each channel and is converted into a reference current signal for the Wilson current mirror (Figure 2), which implies a constant current source topology. Furthermore, through variation of the DAC output signal, the pulse magnitude can be changed over time and, therefore, create different pulse waveforms not limited to rectangular waveforms. This feature allows the system to be used in applications other than neuromuscular stimulation, such as the activation of sensory pathways [28,29]. The components used in the circuit in Figure 2 were designed to offer robustness in power applications.

The current signal reaches an H-bridge circuit that uses four power MOSFETs associated with two high-performance half-bridge driver integrated circuits (Figure 3). This configuration allows the current to flow through the load in both directions and therefore generates differential biphasic pulses, eliminating the fixed anode/cathode polarity of the electrode. Although the system architecture design includes four output circuits, a multiplexing circuit composed of independent solid-state relays allows the stimulation to be carried out by eight channels.

An encoder was used to determine the cycling angle and a belt transmission technique was employed to measure the crank angle. Figure 4 illustrates the designed mechanism, including mechanical drawings of each part. Two pulleys of the same size were connected by a timing belt (model Schneider 2048 MXL-80256 06), in which a driver pulley was associated with the trike crank and a driven pulley was integrated into a rotary incremental encoder (model LPD3806-360BM-G5-24C). The driver pulley featured a rectangular span for zero-crossing detection.

A mobile application (Android) was developed to set and modify the stimulation parameters of the stimulation channels in real time (Figure 5). The muscle groups can be selected along with the start and finish angles at which they are activated. The pulse width, amplitude, and frequency can be modified. The application was designed to collect and save data for offline processing, including the configuration for each channel (ON/OFF, start angle, finish angle, frequency, pulse width, and amplitude), as well as the timestamp, cadence reference value, actual angle, and actual speed. The system exchanges data via the Bluetooth wireless communication protocol at 10 Hz.

Three different modes of operation can be selected through the application: (1) Manual, for Neuromuscular Electrical Stimulation (NMES); (2) Automatic, for FES-assisted cycling, based on simple angle detection; and (3) Control, for FES cycling with cadence control activated. The application permits the pulse parameters to be set and modified during cycling sessions. Global parameter buttons are available in the main screen to increase or decrease the current amplitude, pulse width, and reference cadence (Figure 5a). Individual channel interfaces set the parameters for the right and left sides (Figure 5b). The parameters can be modified even during operation.

### 2.2. Tricycle

Recent research has discussed the methodology and stimulation protocols required for FES-assisted cycling, including the tricycle (trike) mechanical specifications [15,20,27]. Based on their outcomes, we selected a recumbent tadpole trike (Trike Full Suspension, Arttrike, Porto Alegre, RS, Brazil)—fixed on a passive cycling trainer—with two front wheels and a rear wheel, offering greater stability compared to the configuration with one front wheel and two rear wheels. We used a commercially available pedal with calf support (HASE^®^ Bikes, Waltrop, Germany) to keep the ankle joint at 90° and restrict the leg movement to the sagittal plane. A foam layer protected the legs from calf support direct contact. The tricycle and its main components are shown in Figure 6.

### 2.3. Experimental Setup

We designed a single case report in accordance with the recommendations in the CARE guidelines, which state that for case reports to be well-written and transparent, an effective design should reveal early signs of potential benefits, harms, and information in the use of new devices, as well as providing information for clinical research and clinical practice guidelines [30]. The research was approved by a local Ethical Committee (CAAE: 30989620.6.0000.5149, Ethical Approval number 4.190.128), in agreement with the Declaration of Helsinki. 

The stimulation phase for each muscle in the cycle revolution is defined empirically based on the individual’s biomechanics concerning the crank. Therefore, the mechanical optimization of the tricycle (for instance, the adjustment of the crank arm length) must be performed for each subject individually and is crucial in order to obtain a regular cadence. The positioning of the surface electrodes and the definition of a pulse pattern synchronized with the pedaling cycle are also critical. In this study, the vastus medialis (VM), vastus lateralis and rectus femoris (VL + RF) and hamstring (HAM) muscles were activated to perform the cycling movement.

Rectangular self-adhesive surface electrodes (Arktus^®^, Santa Tereza do Oeste, PR, Brazil) of 90 mm × 50 mm were positioned based on previous work [31], as follows: (1) an electrode positioned 2 to 3 cm above the upper edge of the patella on the distal motor point of the vastus medialis muscle, and another electrode on the proximal motor point of the same muscle; (2) an electrode positioned 5 to 7 cm above the upper edge of the patella on the distal motor point of the vastus lateralis muscle, and another electrode on the proximal motor point of the rectus femoris muscle; (3) one electrode placed 5 to 7 cm above the popliteal fossa and the other 15 to 20 cm above the popliteal fossa for hamstring stimulation. The electrode positioning was empirically adjusted to visually obtain the strongest evoked contractions.

### 2.4. Stimulation Pattern

A recent review evaluated ninety-two studies and summarized the characteristics of the stimulation pulses of the intervention protocols as an amplitude of 140 (0–180) mA, frequency of 35 (20–60) Hz, and pulse width of 300 (200–500) µs; the parameters were reported as the median and interquartile range in parentheses [26]. After performance evaluation and adjustments, the final stimulation parameters were set to: 35 Hz frequency, 600 µs pulse width, and up to 100 mA current amplitude. A typical trapezoidal waveform with a ramping up and down with a fixed but configurable percentage of angle range was implemented via firmware to prevent sudden contractions and to achieve more physiological evoked responses. The pulse pattern was validated through a logic analyzer (Saleae Logic Pro 16) at 1.56 MS/s (samples per second). The stimulation pattern for muscle activation is presented in Figure 7.

### 2.5. Control Strategy

In the “Control” operating mode, the current amplitude was adjusted by a proportional-integral (PI) controller tracking a predefined cadence and was limited to 100 mA. Its diagram is illustrated in Figure 8. The cadence error was calculated as the difference between the desired crank angular velocity (reference value) and the measured crank angular velocity.

The control algorithm increases or decreases the pulse amplitude simultaneously for all channels, starting from initial preset values, and was executed by the microcontroller at 10 Hz. This approach was defined to maintain a constant relationship between the activation parameters of each channel/muscle. Although the frequency and pulse width are held constant by the controller algorithm, it is possible to change these parameters manually and for each channel individually through the mobile application, even during operation. Although the pedaling activity assisted by FES is a nonlinear time-varying system, a proportional-integral controller was employed, and the parameters were heuristically defined. The proportional term was empirically adjusted to $K_{p}=k_{1}\cdot 0.45$, where $k_{1}$ was set to a constant value corresponding to 1 mA at the controller output. The integral term was empirically adjusted to reach a minimum error in the steady state and was finally defined as $K_{i}=k_{1}\cdot 0.34$. Initial preset pulse amplitudes were defined as RF + VL = 50 mA, VM = 30 mA, and HAM = 40 mA for the left side, and RF + VL = 45 mA, VM = 25 mA, and HAM = 35 mA for the right side. Equation (1) describes the stimulation control output signal (pulse amplitude):(1)$$u_{x}=p_{x}\left(\theta \right)\cdot h\left(t\right)$$
where $u_{x}$ is the control signal for muscle x={VL+RF,VM, HAM}; *θ* is the crankset position; h(t) is the FES pulse amplitude calculated by the PI control algorithm (Equation (2)); and $p_{x}\left(\theta \right)$ represents the phase control defined by Equation (3):(2)$$h\left(t\right)=K_{p}e\left(t\right)+K_{i}\int_{0}^{t}e\left(\tau \right)d\tau ,\;\forall t\geq 0$$
(3)$$p_{x}\left(\theta \right)=\{\begin{array}{c} 1,\;\;\theta_{x}^{start}\leq \theta \leq \theta_{x}^{end}\\ 0,\;\;otherwise \end{array}$$
where $\theta_{x}^{start}$ and $\theta_{x}^{end}$ correspond to the angle range for the stimulation of muscle x. 

In order to evaluate the controller performance, a methodology for cadence tracking was proposed. A predefined cadence sequence was set, starting at 25 rpm for 60 s, then rising to 38 rpm for 30 s, rising again to 42 rpm and remaining at that level for 60 s before dropping to 38 rpm for 30 s and finishing at 25 rpm for an additional 60 s. The cadence tracking results are illustrated in Figure 9. A moving average filter with a window size of 20 was used for a better visual inspection. The data were collected at 10 Hz. A one-sample Wilcoxon signed-rank test was performed to determine whether the median of the whole pedaling cycle was equal to each of the cadences defined in the protocol.

### 2.6. Subject and Intervention Protocol 

J.S.M., a 68 kg, 38 years old male had suffered from a complete sensory-motor thoracic traumatic SCI (T8; AIS A—American Spinal Injury Association Impairment Scale) [32] for 13 years by the time this work began. He was eligible to participate in the study after eligibility criteria evaluation. His consent to participate was obtained. 

The volunteer participated in lower limb muscle strengthening sessions through NMES stimulation to elicit only isometric contractions for four weeks. The objective was to adapt the musculature of the lower limbs to the FES cycling modality. These sessions lasted for 30 to 45 min and were performed three times a week; the protocol was based on the work presented by Fattal et al. [33]. During each session, the volunteer remained in a sitting position and performed two sub-sessions of 15 min each, one for the stimulation of the rectus femoris and vastus lateralis and gastrocnemius muscles, and another for the hamstrings and vastus medialis muscles. In each of these subsections, the stimulation was alternated between the left and right sides. The pulses were delivered at a sequence of 10 sON/10 sOFF and were applied to the skeletal muscles with an intensity equal to or greater than the motor threshold. The parameters were configured for a pulse width of 500 μs and frequency of 30 Hz. The amplitude was initially adjusted to evoke a visible muscle contraction and was increased automatically by 1 mA after each set of five pulse trains.

After this pre-cycling phase, the subject undertook FES cycling sessions three times a week and the strength training sessions (as with the pre-cycling phase) were kept alternately twice a week. The FES assisted cycling training consisted of an incremental program comprising sessions of a maximum of 45 min, in which the FES control strategy was employed to control the stimulation parameters and increase the duration of the training session. The total distance, time, maximum and average speed were measured using a commercially available speedometer. Incremental load adjustments were performed by switching the bicycle’s gear and through the bicycle’s roller resistance.

From the eleventh month onwards, the volunteer started to practice overground cycling sessions once a week. Although the protocol was not intended to verify whether the volunteer could achieve the Cybathlon 2016 competition requirements [22], we evaluated his performance riding two laps around an athletics track (800 m).

## 3. Results

Bench tests to evaluate the capacity and performance of the system have been discussed in previous work [21]. Nevertheless, Figure 10 evidences the power capacity of the FES device. The pulse measurement was obtained as the potential difference over a 500 Ω load using the oscilloscope’s mathematical subtraction operation (M=ch1−ch2).

Figure 11 illustrates the pulses being triggered in synchronism with the cycling activity for three different channels simultaneously and at different frequencies. The ramping up/down stimulation pattern is also demonstrated.

Regarding the functional outcomes, J.S.M.’s pedaling increased from approximately 20 s to 45 min after the fifth month on the cycling trainer. In this period, in his best performance, he covered 5349 m at an average speed of 7.1 km/h and a maximum speed of 8.0 km/h. From the point at which the experiments were started, pedaling time and distance were evaluated as primary functional parameters related to the load imposed on the pedaling. An improvement in distance and pedaling duration was observed after the cadence control was implemented. The load was adjusted by altering the trike’s gear ratios and through the roller resistance. 

At the end of the eleventh month, as pandemic restrictions eased, it was possible to start cycling overground on an athletic track. During these sessions, pedaling movement was smoothly initiated by the experimenter. J.S.M. was able to pedal six laps around the track (approx. 2.4 km), at an average speed of 4.5 m/h, in 31 min and 15 s. He was also able to complete two laps (800 m) in 495 s. Figure 12 shows J.S.M. on his first ride at the Sports Training Center of the Federal University of Minas Gerais (CTE-UFMG).

The cadence control strategy evidenced that the system was able to follow the predefined cadences. For 25 rpm, the measured reference was 24.84 ± 2.5 (*p* = 0.060); for 38 rpm, the measured reference was 37.86 ± 2.23 (*p* = 0.002); and for 42 rpm, the measured reference was 41.93 ± 2.34 (*p* = 0.208).

## 4. Discussion

The present work discussed the aspects necessary to implement an adaptive FES cycling system and provided more information about the high-capacity stimulation system previously addressed in another context [21]. Our goal was to make this project open-source, including the entire range of hardware, firmware, software, and mechanical designs. With this in mind, we thought that other teams could use this project as a basis to develop other activities assisted by FES. It was also of interest to evaluate the stimulation and control strategies in the FES cycling modality for subjects with SCI and their impact on the overall cycling performance.

After implementing the control strategy using firmware, an increase in distance and pedaling time was observed and the volunteer was able to pedal for 45 min on the passive cycling trainer, a positive outcome considering that other studies have reported shorter durations [15,26]. Regarding the physiological and methodological aspects of the use of electrical stimulation, it is well known that its effectiveness depends less on controllable external factors, such as pulse parameters or the characteristics of the electrodes, and more on the individual’s intrinsic anatomical and physiological properties, which determine the muscle response to electrical stimuli [34]. As the FES cycling system was evaluated with only one volunteer, it is necessary to apply this methodology to other individuals to address which factors lead to a longer pedaling time: whether in fact they are technical aspects or whether they are more related to the responsiveness of the subject. Nevertheless, the proprietary FES cycling system and the methodologies implemented to improve functional electrical stimulation efficiency evidenced the possibility of maximizing pedaling duration for a given individual through control techniques.

The methodology presented proposed the variation of the pulse amplitude while keeping the frequency and pulse width constant. Another study addressed a similar form of cadence tracking using pulse width modulation instead (keeping the frequency and amplitude constant), which evidenced larger cadence oscillation and cadence error compared to our findings [16]. It is unclear, however, whether the results obtained were better due to the way the parameters were varied or whether this was due to other factors, such as trike mechanics, stimulation system performance, subject responsiveness, or the implementation of system control.

A limitation of this study is that the training roller did not allow the consistent setting of the resistance to the pedaling. However, the protocol took place incrementally; that is, as the volunteer managed to pedal at a certain difficulty level, the roller resistance (or the gear ratio) was increased manually through mechanical adjustments to improve the functional aspects (distance) and encourage improvement for the volunteer. Thus, it would be difficult for other researchers to compare the results with the results we obtained in this study. Therefore, considering the distance and time pedaled on the athletic track would be more relevant. It is important to mention, however, that training on the track was limited to six laps and/or 35 min of cycling due to the hot environment and the lack of temperature control. In this case, considering that the Cybathlon competition foresees the completion of 742 m in less than 480 s [22], and taking into account that the tricycle would have been optimized for the competition (weight and tires), not having started the laps from a ramp and not having trained for such activity, we believe we would have achieved a positive result in the competition.

Another constraint is related to the adjustments of the mechanical parameters of the tricycle, which were made for one person only. It was necessary to adapt a commercial trike for the project and it was not possible to modify the seat inclination and crank height. This condition restricted the achievement of the best biomechanical configuration. If other volunteers had participated in the experiment, the mechanical adjustments would have to be carried out individually, which would have demanded a relative effort to define the best FES pulse parametrization (i.e., the stimulation range of each angle). Studies have suggested the use of IMUs as a better approach for cycling phase detection, which could reduce this effort and improve FE -pedaling by SCI individuals [35,36,37].

## 5. Conclusions

The high-power capacity stimulation device was able to generate biphasic electrical pulses with an amplitude, width, and frequency up to 150 mA, 1000 µs, and 100 Hz, respectively. Values higher than those proposed can be tested with this topology to assess the maximum limits. Although the results of the functional tests were obtained from the data from a single case report, the methodology proposed for a better definition of stimulation parameters brought some advantages to the discussion. Our findings suggest a relevant outcome for cadence tracking during FES-assisted cycling sessions for individuals with SCI. Our volunteer was able to cycle for 45 min and cover distances of more than 5 km on a passive cycling trainer as well as pedaling over ground for 35 min. However, it is necessary to evaluate this methodology in a larger population to assess whether these results are repeated for other subjects. The results demonstrated how to implement a cycling tool using a proprietary FES-cycling system and that control techniques can improve FES efficiency.

## Figures and Tables

**Figure 1 sensors-22-00531-f001:**
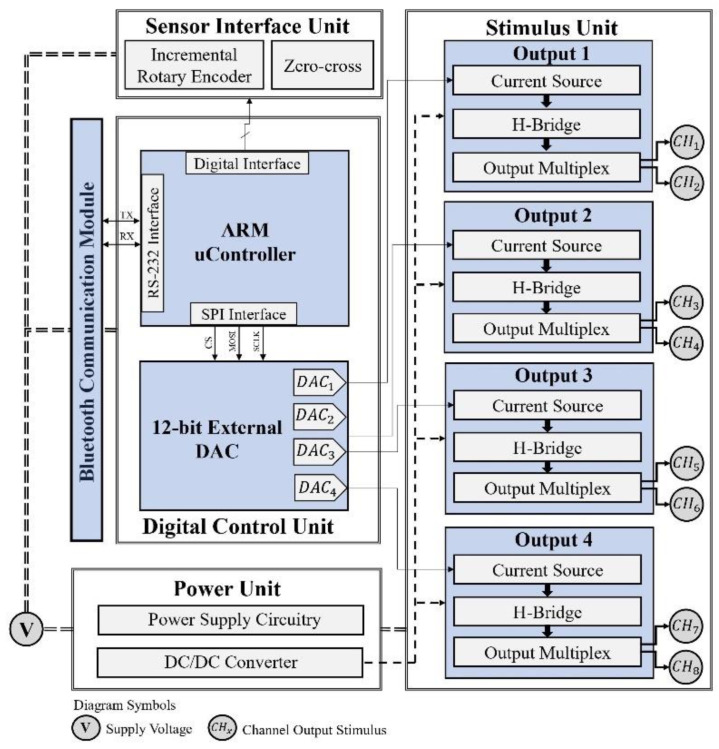
Proprietary 8 channel system architecture composed a digital control unit, responsible for the management of the electrical stimulation system and interface with sensors and communication units; a power supply unit; and a stimulation unit featuring a constant current source topology, which consists of a voltage–current converter, a Wilson current mirror, an H-bridge circuit, and multiplexing circuitry.

**Figure 2 sensors-22-00531-f002:**
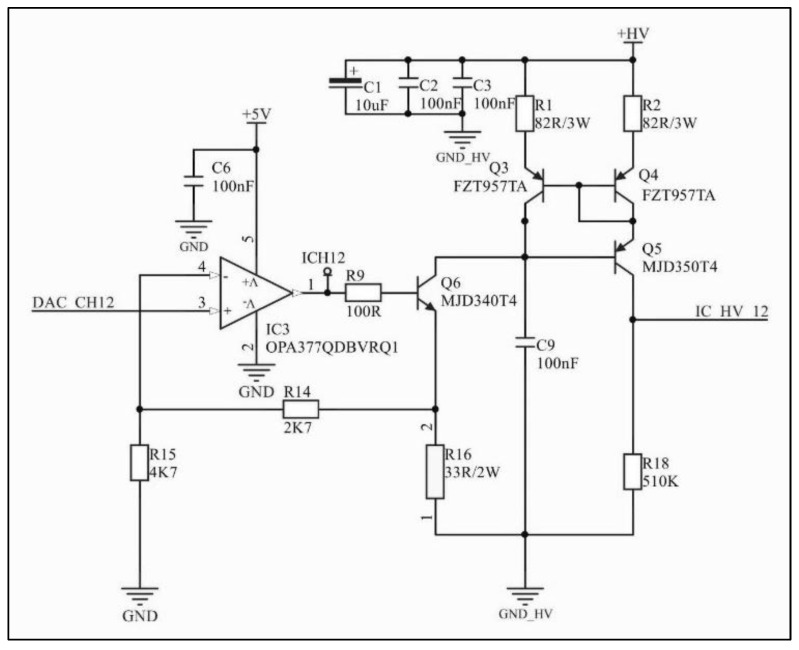
Voltage–current converter and Wilson current mirror. DAC_CH12 is the external DAC output voltage signal; the operational amplifier configuration in association with Q6 and R16 converts the voltage signal into a current signal; the Wilson current mirror controls the current amplitude to remain constant regardless of impedance variations.

**Figure 3 sensors-22-00531-f003:**
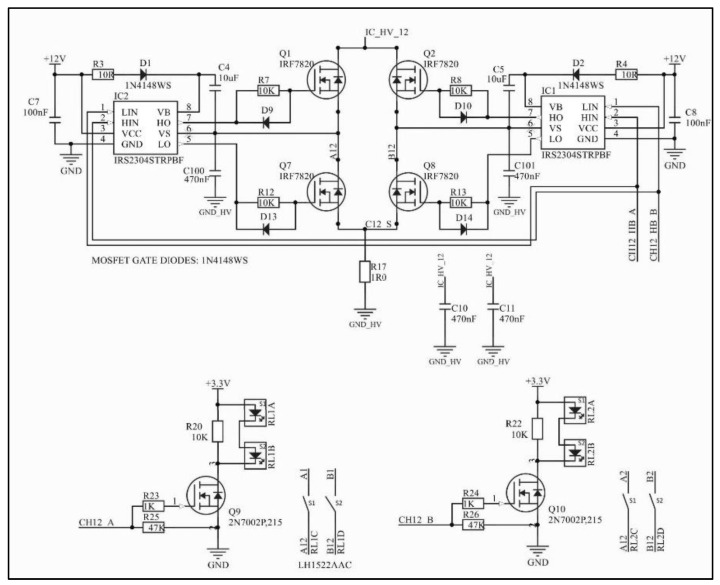
H-bridge circuit and circuitry for activating channels A and B of one of the four independent output channels. The net IC_HV_12 corresponds to the output current signal of the Wilson current mirror (Figure 2). Two independent solid-state relays can be activated, but not simultaneously. Capacitors C100 and C101 are optional.

**Figure 4 sensors-22-00531-f004:**
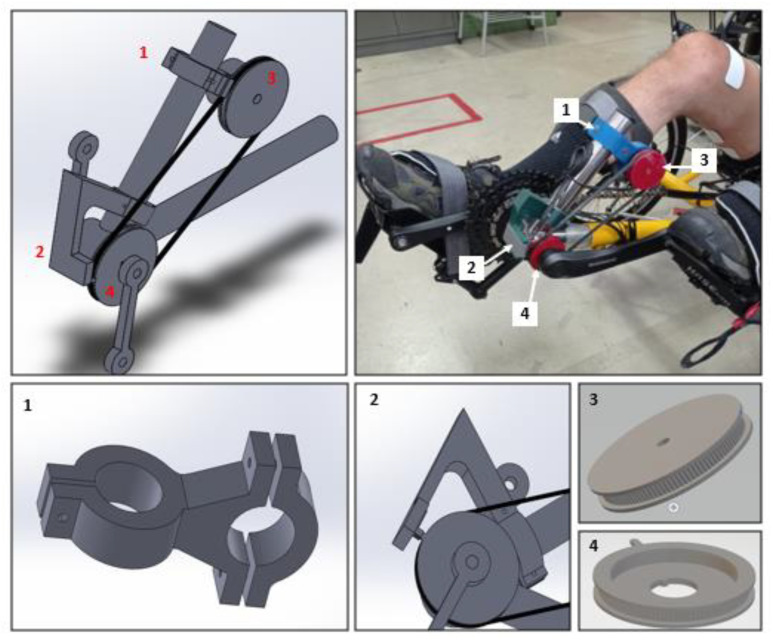
Belt transmission technique to measure the crank angle: (**1**) encoder support part; (**2**) zero-crossing detection part; (**3**) driven pulley attached to the encoder; (**4**) driver pulley with rectangular span connected to the crankshaft.

**Figure 5 sensors-22-00531-f005:**
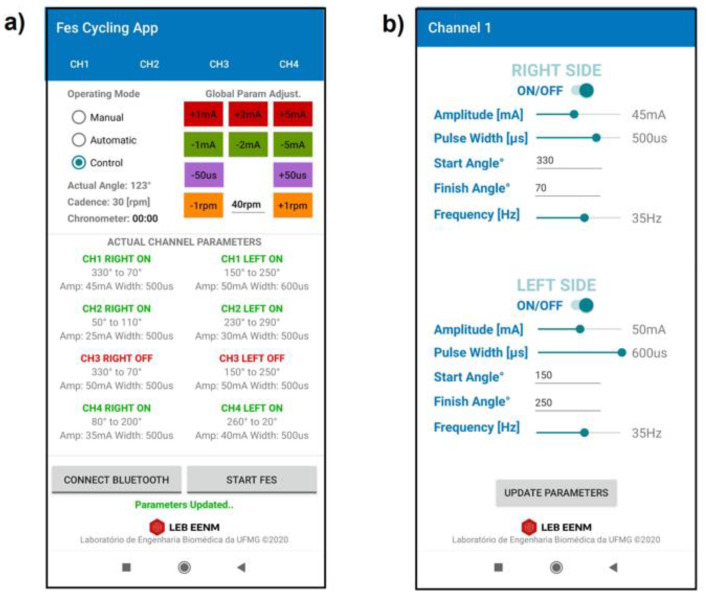
Proprietary mobile application developed for system configuration. (**a**) Main screen displaying real-time data and parameters for *Manual*, *Automatic*, and *Control* modes; (**b**) channel screen used for parameter configuration (channel 1).

**Figure 6 sensors-22-00531-f006:**
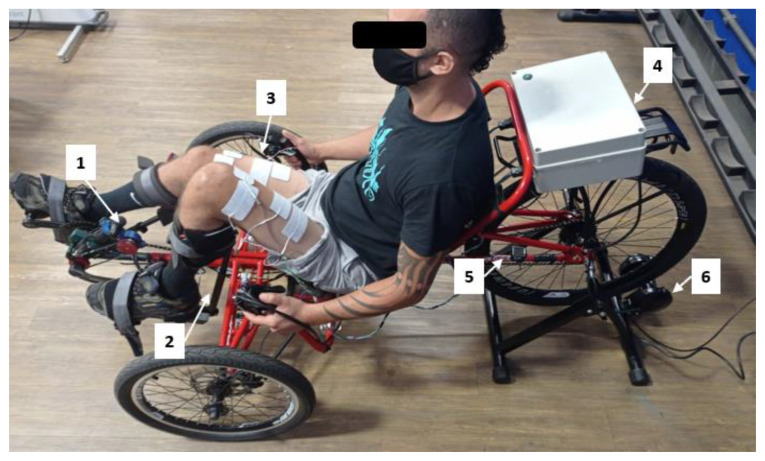
Tadpole tricycle and main components: (**1**) rotary Encoder, (**2**) calf support, (**3**) surface electrodes, (**4**) proprietary stimulator, (**5**) speed sensor, (**6**) cycling trainer.

**Figure 7 sensors-22-00531-f007:**
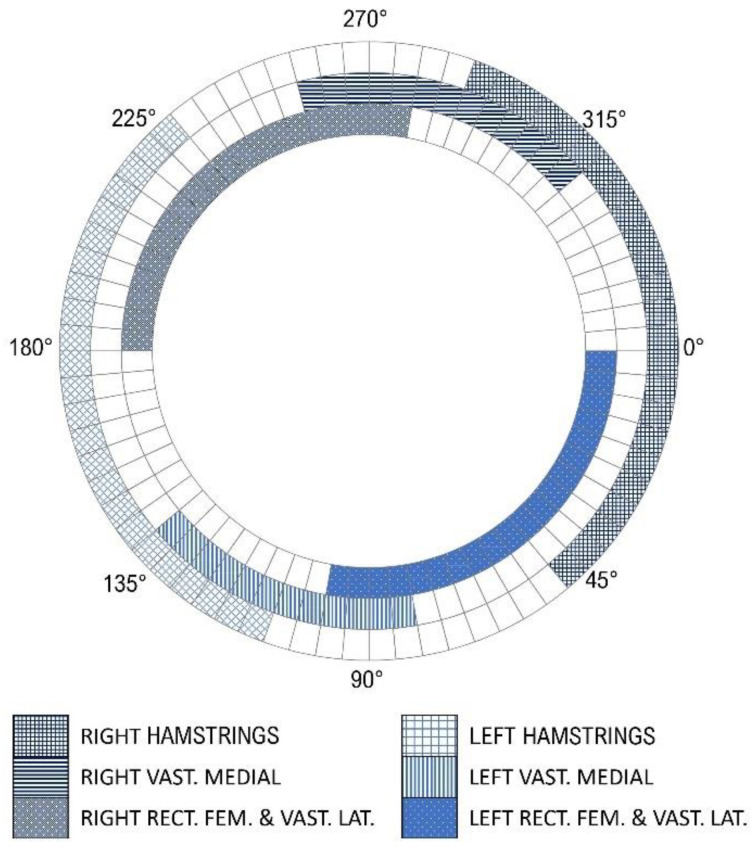
Final stimulation profiles showing the start and end angles for each muscle within a pedaling cycle. Right and left hamstrings (HAM) are shown as the blue grids; right and left vastus medialis (VM) as the horizontal and vertical blue lines, respectively; and vastus lateralis and rectus femoris (VL + RF) as the dotted squares. Zero angle is when the crank arm is positioned horizontally, and the right leg extended.

**Figure 8 sensors-22-00531-f008:**
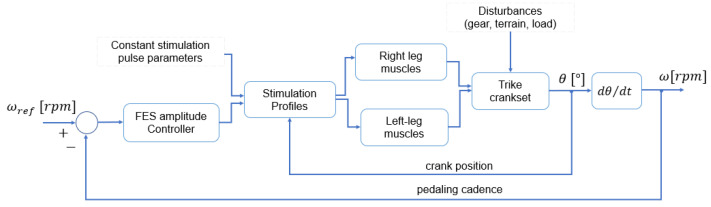
The control strategy implemented for FES cycling cadence tracking.

**Figure 9 sensors-22-00531-f009:**
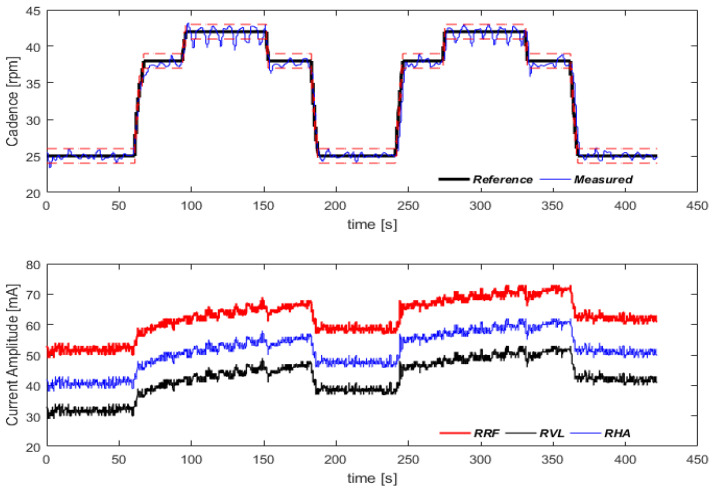
Cadence tracking protocol. In the first graph, the dashed lines around the reference cadence correspond to ±1 rpm. Variation in pulse amplitude for the three muscles of the right lower limb are shown in the second graph. Video for demonstration: https://youtu.be/6cgDhPWPzII (accessed on 21 October 2021).

**Figure 10 sensors-22-00531-f010:**
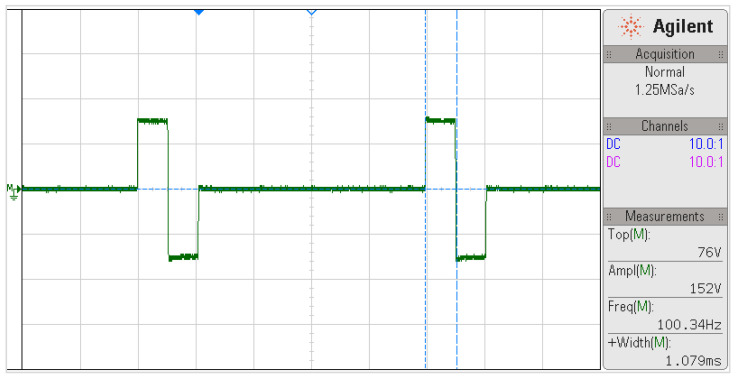
Biphasic pulse waveform at 100 Hz frequency, 1000 µs width, and 150 mA current amplitude for a 500 Ω load. The pulse measurements are shown in the bottom right field.

**Figure 11 sensors-22-00531-f011:**
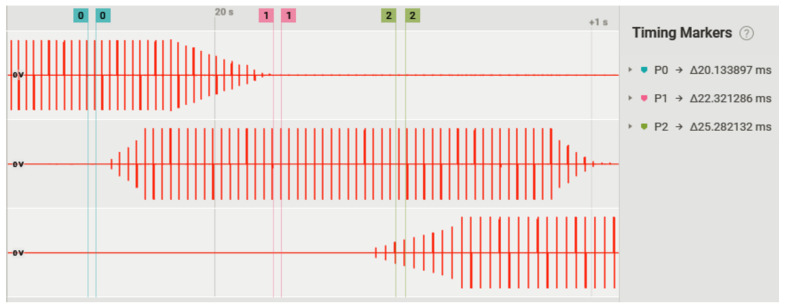
Triggering of biphasic rectangular pulses for three different channels simultaneously and in different segments of the pedaling cycle. Channel 1 (upper) at 50 Hz; channel 2 (in the middle) at 45 Hz; channel 3 (lower) at 40 Hz.

**Figure 12 sensors-22-00531-f012:**
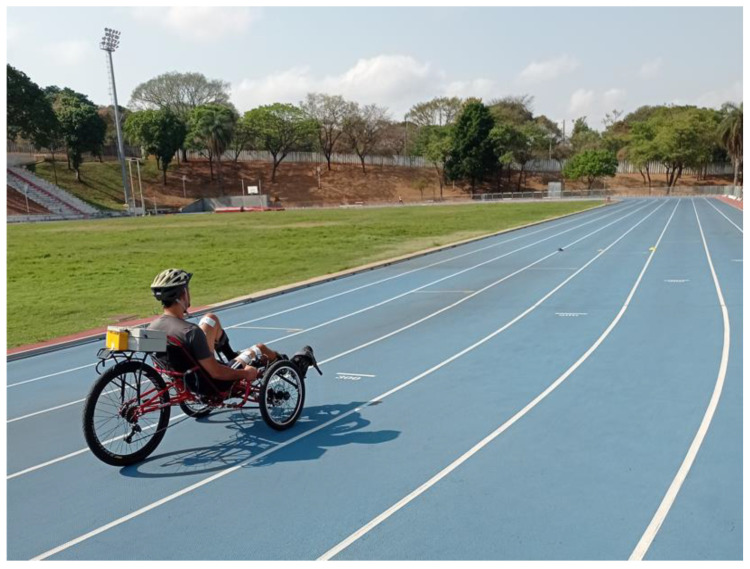
First FES-assisted cycling session on the athletics track of the UFMG Sports Training Center (CTE-UFMG). Video for demonstration: https://youtu.be/J9IXGMNagf0 (accessed on 20 August 2021).

## Data Availability

Data sharing is not applicable to this article.

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
