# Peer review of "Development of a High-Power Capacity Open Source Electrical Stimulation System to Enhance Research into FES-Assisted Devices: Validation of FES Cycling"

_sensors, 2022, doi:10.3390/s22020531_

Round 1

Reviewer 1 Report

In this study, the The high-power capacity stimulation device was developed and employed to generate biphasic electrical pulses with 150mA [amplitude], 1000us [w]idth] and 100Hz [frequency]. The authors address the important possibility to apply for SCI patient.  the results show how to implement FES-cycling system with control techniques. I think that It is impressive works.

Author Response

Thank you for the feedback. No further comments have to be addressed.

Reviewer 2 Report

This paper presents an eight channel FES system configured to practice FES-Cycling in people with SCI. The system was proprietary and now they present this systems as an open-source system.

For this reviewer, this topology of stimulator is on state-of-art of stimulators and I can´t see any new contribution for the topic. They only present the stimulator validated on one person, but they not indicate the effect of the stimulation and other effects on the person, for example, the waveform presented on Fig. 10 produce a high pain when stimulus are applied.

Review the figure 10, because you present a "math" signal in red but is not possible to see the channel 2 signal in blue-green. How is calculated the signal Math ? 

The paper is well developed but the scientific significance is low, because I think it not generates any new knowlegede for the FES systems development. The authors are recognized as high level researchers on the topic.

Reviewer 3 Report

This manuscript described a multichannel functional electrical stimulation system. The system can be used for passive cycling and was tested on a participant with a spinal cord injury.

The manuscript is concise and well written. The question is whether the work is novel given that passive cycling systems exists, but the objective is to make their system public with some code available at https://osf.io/pf3ru/. 

Lines 209 - 217: It was not clear how motor point of the muscles were identified. 

Lines 216 - 217: Were electrodes place at the motor point and later empirically moved?

Figure 7 requires a better explanation. .

lines 242 -243: Is that meant to say all channels on one leg?

Lines 248 - 249 : Author could expand on the system identification procedure that led to  Kp and Ki.

Lines 290 - 299: It is not clear how the FES was controlled when the participant was using the system. How did FES assist the participant? What was the FES control strategy during the FES-assisted cycling training sessions? These needed to be made clearer

Figure 10 could be presented better.
